# Relationships between Gait Regularity and Cognitive Function, including Cognitive Domains and Mild Cognitive Impairment, in Community-Dwelling Older People

**DOI:** 10.3390/healthcare9111571

**Published:** 2021-11-18

**Authors:** Takasuke Miyazaki, Ryoji Kiyama, Yuki Nakai, Masayuki Kawada, Yasufumi Takeshita, Sota Araki, Hyuma Makizako

**Affiliations:** 1Department of Physical Therapy, School of Health Sciences, Faculty of Medicine, Kagoshima University, Kagoshima 891-0175, Japan; k5588736@kadai.jp (T.M.); y-nakai@daiichi-koudai.ac.jp (Y.N.); kawada@health.nop.kagoshima-u.ac.jp (M.K.); k1740552@kadai.jp (S.A.); makizako@health.nop.kagoshima-u.ac.jp (H.M.); 2Department of Rehabilitation, Tarumizu Municipal Medical Center, Tarumizu Central Hospital, Kagoshima 891-2124, Japan; k2687318@kadai.jp; 3Department of Mechanical Systems Engineering, Daiichi Institute of Technology, Kagoshima 899-4395, Japan; 4Graduate School of Health Sciences, Kagoshima University, Kagoshima 891-0175, Japan

**Keywords:** cognition, wearable sensor, acceleration, gait analysis, aging

## Abstract

The aim of this cross-sectional study was to examine the correlations between gait regularity, cognitive functions including cognitive domains, and the mild cognitive impairment (MCI) in community-dwelling older people. This study included 463 older adults (63.4% women, mean age: 74.1), and their step and stride regularity along the three-axis components was estimated from trunk acceleration, which was measured by inertial measurement units during a comfortable gait. Four aspects of cognitive function were assessed using a tablet computer: attention, executive function, processing speed, and memory, and participants were classified into those with or without MCI. The vertical component of stride and step regularity was associated with attention and executive function (r = −0.176–−0.109, *p* ≤ 0.019), and processing speed (r = 0.152, *p* < 0.001), after it was adjusted for age and gait speed. The low vertical component of step regularity was related to the MCI after it was adjusted for covariates (OR 0.019; *p* = 0.016). The results revealed that cognitive function could affect gait regularity, and the vertical component of gait regularity, as measured by a wearable sensor, could play an important role in investigating cognitive decline in older people.

## 1. Introduction

Gait and cognition are known to be closely correlated [1], and gait impairment is a risk factor for people with mild cognitive impairment (MCI) and dementia [2,3,4]. MCI is a transitional state of cognitive loss, or results in lower cognitive performance, specifically attention, executive function, processing speed, and memory domain, than that of which is expected for a given age group or educational level, but does not yet fulfill the clinical criteria for probable Alzheimer’s disease [5]. MCI is characterized by a heterogeneous decline in one or more cognitive domains [6]. MCI is a precious window of opportunity for the prevention of dementia. Previous studies show that slow gait and high gait variability occurs in older people with MCI [1,7,8,9], and that slow gait with MCI can lead to adverse events such as falls and disability [8]. Therefore, it seems important to conduct gait assessment in older people when considering the early stages of cognitive decline such as MCI, in order to maintain gait and cognitive function.

Gait is usually assessed by various parameters, with gait speed being one of the most common gait parameters [10,11]. In addition, a previous report indicates that gait variability, including step time variability, is a better predictor of cognitive decline than gait speed in cognitively normal older adults [12]. Therefore, gait variability could also be an important indicator for older people with cognitive decline.

Gait regularity, which reflects an aspect of gait variability, is often measured using wearable sensors and a walkway with embedded pressure sensors [13,14,15]. Moreover, gait regularity has been associated with decreased aging [16], and is related to falls [17] and cognitive decline [18,19]. Wearable sensors have proved to be useful during gait analysis as they permit a simple, objective assessment of human gait. Therefore, we suggest that the assessment of gait regularity using wearable sensors might be useful for investigating cognitive decline in older people.

Previous studies have shown that gait parameters that are measured by wearable sensors, including stride length and stride time variability, relate to the memory, executive, and attention functions in community-dwelling older people [20,21,22]. However, the correlation between gait parameters and cognitive function, including cognitive domains and severity, has not been fully clarified due to the lack of large samples in previous studies. Gait measurements using wearable sensors in a cohort field would provide useful information concerning the relationship between cognition and gait. Therefore, the aim of this study was to examine the correlations between gait regularity, as measured by wearable sensors, cognitive functions including cognitive domains, and MCI in community-dwelling older people. We hypothesized that gait regularity correlates with several cognitive domains and MCI. This information provides further insight into gait assessment while taking account of cognitive function in older people.

## 2. Materials and Methods

### 2.1. Participants

The present cross-sectional study used data from the Tarumizu Study 2018, which was implemented in cooperation with Kagoshima University (Faculty of Medicine), Tarumizu City Office, and Tarumizu Chuo Hospital, and was held between June and December 2018 as a community-based health check survey. The individuals selected to participate in the Tarumizu Study 2018 were chosen from among the older communities living in Tarumizu City, near Kagoshima, Japan. Participants were recruited through local newspapers and campaigns in community events. The inclusion criteria were living in Tarumizu City, aged ≥65, and able to walk without walking aids. The exclusion criteria were a specific medical history of neurological and orthopedic diseases, Parkinson’s disease, dementia, depression, fractures, osteoarthritis, severe visual or auditory impairment, cognitive impairment (Mini-cog ≤ 2) [23,24], receiving support from the Japanese public long-term-care insurance system, and/or missing data. As a result of recruitment, 463 people completed the neuropsychological assessments and gait experiments (Figure 1).

The participants’ basic information including age, sex, and body mass index were recorded. In addition, medical conditions, including current medications, medical history, and educational history were recorded using a questionnaire. Informed consent was obtained from all participants before their inclusion in the study, and the Ethics Committee of the Faculty of Medicine, Kagoshima University approved the study protocol (ref no. 170103).

### 2.2. Gait Measurement

Participants walked twice at a comfortable velocity along a 14 m straight walkway, and we calculated gait regularity from the acceleration of the pelvis during gait. This acceleration was measured using a magnetic inertial measurement unit (MIMU) (Mtw Awinda, Xsens, Enschede, NL) with a sampling rate of 100 Hz. The MIMU consisted of a 3D rate gyroscope, 3D accelerometer, and 3D magnetometer, and we measured trunk three-axis acceleration along the anterior-posterior (AP), medial-lateral (ML), and vertical (VT) lines in the global coordinate system. The MIMU device was fixed at the posterior of the sacrum, and was calibrated so that the vertical direction of the coordinate system was along the gravity in static standing [25].

In this study, we calculated stride regularity and step regularity along three axes; a detailed protocol of which has been previously described [14]. Step regularity was assessed using the autocorrelation function of trunk acceleration at the average step time lag, while stride regularity was estimated from acceleration by an autocorrelation function at the average stride time lag, using the Pearson product-moment correlation coefficient. These values ranged from −1 to 1, and when similarity of the stride or step was high, the value was close to 1. We estimated gait regularity from trunk acceleration in the middle of five walking cycles, and the mean of 10 strides from two passages at a comfortable gait; this was adopted as the representative value [14]. Data processing was performed using MATLAB R2017b (Mathworks Inc., Natick, MA, USA) mathematical software.

### 2.3. Cognitive Assessment

Cognitive domains were assessed using the National Center for Geriatrics and Gerontology Functional Assessment Tool (NCGG-FAT) [26]. The NCGG-FAT consists of assessments of multiple cognitive domains including attention, executive function, processing speed, and memory. Attention and executive function were assessed using the electronic tablet version of the Trail Making Test, parts A and B [26]. Processing speed was assessed using a tablet version of the Symbol Digit Substitution Task [26], based on the Symbol Digit Modalities Test. This test provides nine pairs of numbers and symbols, and participants select the number corresponding to a target symbol. The score is the number of correct answers chosen within 90 s. We recorded the amount of time it took to complete each task. Memory was assessed using the subtest of Alzheimer’s Disease Assessment Scale‒cognitive, including immediate recognition [27]. Better performance is represented by lower values in attention, executive function, and higher values in the other tests.

### 2.4. Mild Cognitive Impairment

MCI criteria were established and revised by Petersen [5], so that participants satisfy the following conditions: (i) subjective cognitive complaints; (ii) objective cognitive decline; (iii) intact general cognitive function; and (iv) function independently in daily activities. MCI was diagnosed using the NCGG-FAT [26], and this detailed protocol is described in a past study [28]. All tests used in this study had standardized thresholds for the definition of MCI based on objective cognitive impairment (score < 1.5 SDs below age- and education-specific means), based on a previous cohort database of community-dwelling older Japanese people [28].

### 2.5. Statistical Analysis

The unpaired *t*-test for continuous variables or chi-square test for categorical variables were conducted to examine the differences in participant characteristics, including basic information, gait regularity, and cognitive domains between MCI and non-MCI. Furthermore, partial correlations, with the control of the effect of age and gait velocity, were calculated to determine the relationships between gait regularity and cognitive domains. Finally, binomial logistic regression analysis (adjusted for age, sex, body mass index (BMI), history of education, medications, and gait velocity) was conducted to explore the association between MCI (dependent variable) and gait regularity (independent variable). All statistical analyses were performed using SPSS 25 (IBM, Armonk, NY, USA), and the significance level was set at 5%.

## 3. Results

### 3.1. Characteristics of the Participants According to MCI Status

The 463 participants (63.4% women, mean age: 74.1) completed the cognitive assessments and gait experiments, and they were classified as non-MCI (*n* = 342) or MCI (*n* = 121) (Table 1 and Table 2). The MCI group had a greater age (*p* < 0.001), a slower gait speed (*p* = 0.001), and they scored lower in all cognitive domains (*p* < 0.001; Table 1).

In terms of gait regularity, the VT component of stride regularity (*p* = 0.017), AP component of step regularity (*p* = 0.007), and VT component of step regularity (*p* < 0.001) were decreased in the MCI group (Table 2).

### 3.2. Association between Gait Regularity and Cognitive Functions

In a partial correlation analysis that was adjusted for age and gait velocity, low stride regularity-VT was associated with the decrease in attention function (r = −0.139; *p* = 0.003) and executive function (r = −0.109; *p* = 0.019, Table 3). Furthermore, low step regularity-VT was associated with the decrease in attention function (r = −0.165; *p* < 0.001), executive function (r = −0.176; *p* < 0.001), and processing speed (r = 0.152; *p* < 0.001; Table 3). Decreased VT components of stride and step regularity were related to several decreased cognitive domains.

In addition, logistic regression analysis showed that low step regularity-VT was related to the MCI (odd ratio 0.019; 95% confidence interval 0.001–0.473; *p* = 0.016) after adjustments for age, sex, BMI, history of education, medications, and gait velocity (Table 4).

## 4. Discussion

In this study, correlations were observed between gait regularity (measured using an MIMU), cognitive functions (including cognitive domains), and the presence of MCI in community-dwelling older people. The results showed that the MCI group exhibited a decreased gait regularity, a reduction in all cognitive domains, and a lower VT component of stride and step regularity, all of which correlated to the decrease in several cognitive domains. In addition, the VT component of step regularity was particularly associated with the MCI. Therefore, the results of this study suggest that the vertical regularity during gait, as estimated by an MIMU, could reflect the cognitive function and MCI in older people.

In this study, the MCI group exhibited a lower stride and step regularity than the non-MCI group. Previous studies similarly showed a high variability in stride and swing time in older people with MCI or dementia than in healthy subjects [9,15]. Furthermore, even excluding the effects of age and gait velocity, a partial correlation analysis showed that low stride regularity-VT was associated with a decrease in attention and executive function, and low step regularity-VT was associated with a decrease in attention, executive function, and processing speed. These relations were consistent with the findings of previous studies, which reported that step time variability is correlated with attention in patients with dementia [20], and executive function in community-dwelling older people [29]. A previous study also described that variability during the walking that occurs while participants are responding to a cognitive task is sensitive to deficits in executive function and processing speed, as these cognitive tasks diminish the central processing resources that are required during the natural walking [30]. This issue demonstrates the close interaction between cognitive function and walking function. Therefore, gait regularity calculated from trunk acceleration reflects the cognitive function and would be a useful gait parameter for gait assessment using an MIMU when investigating attention, executive function, and processing speed, in older people.

In addition, logistic regression analysis showed that step regularity-VT was particularly related to MCI. A previous study shows that an MCI group had a lower step regularity-VT than a non-MCI group [13]. Meanwhile, stride regularity of the VT component, as estimated by a harmonic ratio from trunk acceleration, has been correlated with falls in community-dwelling older people [31]. The current results are consistent with these reports, and the VT component of gait regularity would be one of the important gait parameters associated with decreased gait function and cognitive decline in older people. Therefore, this study suggests that the VT component of step regularity calculated from trunk acceleration, as measured by an MIMU, may be an especially useful indicator for cognitive decline, such as MCI, in older people.

One of the most notable strengths of this study is that the sample size of the cohort for community-dwelling older people depends on the accuracy and ease of use of an MIMU [13]. Furthermore, this study showed that cognitive domains and the severity of cognition could affect gait regularity, as measured by an MIMU, when adjusted for age and gait speed. Therefore, it is suggested that gait analysis using an MIMU should be made more widely available in communities of older people. Moreover, gait regularity calculated from trunk acceleration, as measured by an MIMU, could play an important role in identifying cognitive decline in older people.

However, several limitations must be noted. As a cross-sectional design was used, the causal relationships between cognitive function and gait regularity were still unclear in community-dwelling older people. Further prospective studies are required to address this issue. We did not consider subtypes of MCI owing to there being insufficient samples for classification into subtypes. Recently, the relationships between gait and brain function were analyzed to prevent cognitive decline [32,33,34,35]. Further studies from various perspectives are needed to clarify the causal relationships between cognitive function and gait regularity in participants with cognitive impairment.

## 5. Conclusions

Gait regularity was influenced by cognitive function, including attention, executive function, and processing speed. Meanwhile, the low VT component of step regularity was also influenced by the decreased cognitive function and MCI. This study suggested that the VT component of gait regularity, estimated by MIMU, could play an important role in investigating cognitive decline, such as MCI in older people. Further studies are warranted to clarify the causal relationships between cognitive function and gait regularity in participants with cognitive impairment. These attempts might contribute to early detection for the prevention of dementia using mobile sensing.

## Figures and Tables

**Figure 1 healthcare-09-01571-f001:**
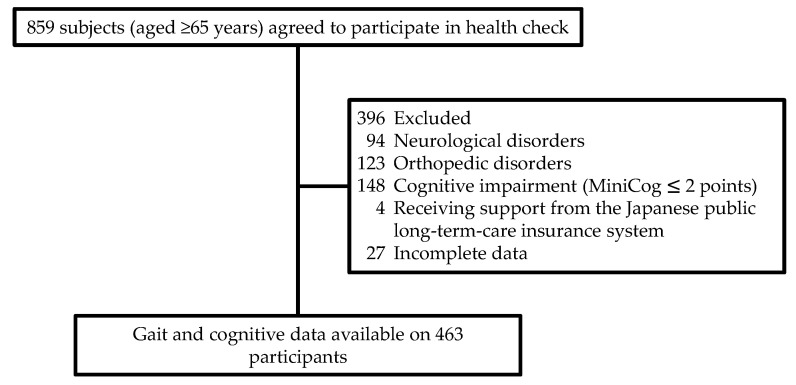
Subject flow diagram from initial contact through to study completion.

**Table 1 healthcare-09-01571-t001:** Characteristics of the demographic data including cognitive functions.

Variables	Overall (*n* = 463)	No MCI (*n* = 342)	MCI (*n* = 121)	*p*-Value
Age (year)	74.1 ± 5.8	73.5 ± 5.6	75.6 ± 6.0	<0.001
Female, *n* (%)	294 (63.4)	217 (63.4)	77 (63.6)	0.971
Height (cm)	155.0 ± 8.3	155.2 ± 8.2	154.4 ± 8.7	0.371
Weight (kg)	55.8 ± 9.8	56.1 ± 9.8	54.7 ± 9.7	0.170
Body mass index	23.1 ± 3.2	23.2 ± 3.1	22.9 ± 3.2	0.310
Current medication	2.73 ± 2.7	2.65 ± 2.6	2.97 ± 2.9	0.263
Educational history (year)	11.4 ± 2.2	11.5 ± 2.2	10.9 ± 2.2	0.009
Gait speed (m/s)	1.30 ± 0.2	1.32 ± 0.2	1.24 ± 0.2	0.001
Cognitive functions				
Attention (s)	22.9 ± 8.1	20.4 ± 4.7	29.9 ± 11.2	<0.001
Executive (s)	47.7 ± 31.7	36.9 ± 11.9	78.2 ± 46.9	<0.001
Process speed (score)	41.4 ± 10.6	44.2 ± 9.3	33.5 ± 10.3	<0.001
Memory (score)	7.56 ± 1.4	7.86 ± 1.1	6.71 ± 1.6	<0.001

Values are expressed as mean ± SD. MCI, Mild Cognitive Impairment.

**Table 2 healthcare-09-01571-t002:** Characteristics of gait regularity.

Component	Overall (*n* = 463)	no MCI (*n* = 342)	MCI (*n* = 121)	*p*-Value
Stride regularity				
Anteroposterior	0.84 ± 0.09	0.84 ± 0.09	0.83 ± 0.09	0.111
Mediolateral	0.66 ± 0.13	0.67 ± 0.13	0.65 ± 0.14	0.127
Vertical	0.82 ± 0.09	0.83 ± 0.09	0.80 ± 0.11	0.017
Step regularity				
Anteroposterior	0.81 ± 0.09	0.82 ± 0.09	0.80 ± 0.09	0.007
Mediolateral	−0.56 ± 0.14	−0.58 ± 0.14	−0.54 ± 0.15	0.058
Vertical	0.77 ± 0.11	0.78 ± 0.10	0.73 ± 0.12	<0.001

Values are expressed as mean ± SD. MCI, Mild Cognitive Impairment.

**Table 3 healthcare-09-01571-t003:** Partial correlations after controlling the effect of age and gait velocity among gait regularity and cognitive domain function (*n* = 463).

Component	Attention	Executive Function	Processing Speed	Memory
Stride regularity				
Anteroposterior	−0.060	−0.036	−0.007	0.023
Mediolateral	−0.068	−0.016	0.015	0.026
Vertical	−0.139 *	−0.109 *	0.006	−0.003
Step regularity				
Anteroposterior	−0.094 *	−0.066	0.073	0.056
Mediolateral	0.078	0.059	0.073	0.056
Vertical	−0.165 **	−0.176 **	0.152 **	0.084

* *p* < 0.05., ** *p* < 0.01.

**Table 4 healthcare-09-01571-t004:** Binomial logistic regression analysis of the relationship between MCI and gait regularity after adjusting for covariates (*n* = 463). The covariates included age, sex, BMI, history of education, medications, and gait velocity.

Component	Odds Ratio (95% CI)	*p*-Value
Stride regularity		
Anteroposterior	0.365 (0.015–8.827)	0.535
Mediolateral	2.273 (0.012–435.6)	0.759
Vertical	2.496 (0.033–189.5)	0.679
Step regularity		
Anteroposterior	0.684 (0.081–5.753)	0.726
Mediolateral	0.728 (0.015–36.49)	0.874
Vertical	0.019 (0.001–0.473)	0.016

## Data Availability

The data used to support the findings of current study are available from the corresponding author upon request.

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
