# Peer review of "Relationships between Gait Regularity and Cognitive Function, including Cognitive Domains and Mild Cognitive Impairment, in Community-Dwelling Older People"

_healthcare, 2021, doi:10.3390/healthcare9111571_

Round 1

Reviewer 1 Report

This paper is well written and the objectives, methods and aims are clearly stated.

It does add some clarity around the components of executive function that best correlate with gait variability.

I have some minor comments:

  • I needed to read reference 13 to fully understand the method and obtain meaning of the variables in Table 2 and 3 relating to “stride and step regularity”. I suggest expanding the paragraph between lines 98-105 to make the origins and units of these parameters a little clearer.
  • Correlations whose r value<0.2 are not very strong and the p values can easily be significant in populations this large. The reported r values are only contributing to quite a small amount of the variance. I suggest that the authors temper their wording and not overcall the relationships.

Reviewer 2 Report

The manuscript describes the associations of gait regularity with cognitive domains, as well as MCI in community-dwelling older adults. Strengths of this study/manuscript are the large sample size, clear study rationale, well-described methodology allowing for reproducibility, clearly presented study results, and appropriate conclusions. Overall the manuscript is well-written, confirms previous and reveal new knowledge to the research field of gait and cognition. I do have only few requests the authors should address.

Major comments:

  • Given the title of the manuscript, I was surprised to read about the exclusion criterion of cognitive impairment (MiniCog<3). Can the authors comment on that? Why was this exclusion criterion used? I assume that the analyses presented in the manuscript cover secondary analyses from the Tarumizu Study 2018, in which this criterion was defined?! So there has also been no a prior sample size calculation for the presented analyses.
  • The correlation coefficients for the cognitive domains with gait regularity are quite small (all r<.20), but significant due to the large sample size. The authors mentioned that these results are consistent with previous studies (line 184-186). What magnitude of associations are reported in these studies?
  • The argument in lines 186-192 is not totally clear to me and should be specified.
  • Do the authors have some ideas why the VT component of gait regularity is related to cognitive domains and MCI, but Not the AT and ML components? Did other studies investigate associations of cognitive domains with trunk accelerations in different directions?

Minor comments:

  • Line 19: „…were estimated from…“
  • Line 116: Why results were excluded from analysis if the time was greater than 90s?
  • Table 2: Please provide a unit for gait regularity.

Reviewer 3 Report

Summary: The current manuscript aims to evaluate the relationship Relationships between gait regularity and cognitive function, also considering MCI in older people. The authors show that cognitive function would affect gait regularity, and the vertical component of gait regularity as measured by a wearable sensor could play an important role in investigating cognitive decline in older people. Although authors present interesting findings, some aspects could be improved.

Introduction: Overall, the introduction provides a broad background and rationale for the research. However, some aspects about cognitive functions in MCi could be improved (for a review see Guarino, A., Forte, G., Giovannoli, J., & Casagrande, M. (2020). Executive functions in the elderly with mild cognitive impairment: a systematic review on motor and cognitive inhibition, conflict control and cognitive flexibility. Aging & mental health, 24(7), 1028-1045.)  

Methods: The method are comprehensive

Analysis: the analyses are well conducted.

Results: The summary of the study provided is well-defined and fits according to the analysis plan provided.

Discussion: the conclusions appear to be a summary of the results, I suggest reporting the usefulness of this study and further perspective.

General comment: I would also encourage the authors to check all references.
